# Clinicopathologic Significance of Heat Shock Protein 60 as a Survival Predictor in Colorectal Cancer

**DOI:** 10.3390/cancers15164052

**Published:** 2023-08-11

**Authors:** Myunghee Kang, Soyeon Jeong, Jungsuk An, Sungjin Park, Seungyoon Nam, Kwang An Kwon, Debashis Sahoo, Pradipta Ghosh, Jung Ho Kim

**Affiliations:** 1Department of Pathology, Gachon University Gil Medical Center, College of Medicine, Gachon University, Incheon 21565, Republic of Korea; kangmh@gilhospital.com; 2Gachon Medical Research Institute, Gachon Biomedical Convergence Institute, Gachon University Gil Medical Center, College of Medicine, Gachon University, Incheon 21565, Republic of Korea; jensyj85@gmail.com; 3Department of Pathology, Korea University Anam Hospital, College of Medicine, Korea University, Seoul 02841, Republic of Korea; anbox@naver.com; 4Department of Genome Medicine and Science, AI Convergence Center for Medical Science, Gachon University Gil Medical Center, Gachon University College of Medicine, Incheon 21565, Republic of Korea; oscar.park@gmail.com (S.P.); nams@gilhospital.com (S.N.); 5Department of Health Sciences and Technology, Gachon Advanced Institute for Health Sciences and Technology (GAIHST), Gachon University, Incheon 21565, Republic of Korea; 6Department of Internal Medicine, Gachon University Gil Medical Center, College of Medicine, Gachon University, Incheon 21565, Republic of Korea; toptom@gilhospital.com; 7Department of Computer Science and Engineering, University of California, San Diego, CA 92093, USA; dsahoo@health.ucsd.edu; 8Department of Pediatrics, University of California, San Diego, CA 92093, USA; 9Department of Cellular and Molecular Medicine, University of California, San Diego, CA 92093, USA; prghosh@ucsd.edu; 10Department of Medicine, University of California, San Diego, CA 92093, USA; 11HUMANOID Center of Research Excellence (CoRE), University of California, San Diego, CA 92093, USA

**Keywords:** colorectal cancer, heat shock protein 60, heat shock protein family D (HSP60) member 1, TNM classification

## Abstract

**Simple Summary:**

HSP60, a mitochondrial chaperone, can promote or inhibit cancer progression. Patients with colorectal cancer (CRC) were examined for HSP60 expression using the TNM classification. Patients with differentiated and p53-mutated CRC expressed high levels of HSP60. Compared with patients with high HSP60 expression, those with low expression had event-free and disease-specific survival hazard ratios of 1.42 and 1.69, respectively. TNM class and HSP60 expression affected survival, especially in late/advanced stages. The expression of the *HSPD1* gene, which encodes HSP60, exhibited the same pattern as the protein. The hazard ratios for overall and relapse-free survival were 1.80 and 1.87, respectively, for patients with reduced *HSPD1* expression. Low *HSPD1* expression and advanced malignancy worsen CRC prognosis. This study suggests that *HSPD1*/HSP60 may be a useful biomarker for refined survival prediction in late-stage and advanced-stage CRC, allowing for individualized therapy.

**Abstract:**

The role of heat shock protein 60 (HSP60), a mitochondrial chaperone, in tumor progression or its anti-tumor effects remains controversial. This study aimed to confirm the possibility of using HSP60 as a prognostic marker in patients with colorectal cancer (CRC), considering TNM classification for precise prediction. HSP60 expression increased with differentiation and p53 mutations in patients. However, compared to patients with high HSP60 expression, patients with low HSP60 expression had event-free survival and disease-specific survival hazard ratios (HRs) of 1.42 and 1.69, respectively. Moreover, when the survival rate was analyzed by combining TNM classification and HSP60 expression, the prognosis was poor, particularly when HSP60 expression was low in the late/advanced stage. This pattern was also observed with HSP family D member 1, *HSPD1*, the gene that encodes HSP60. Low *HSPD1* expression was linked to lower overall survival and relapse-free survival rates, with HRs of 1.80 and 1.87, respectively. When TNM classification and *HSPD1* expression were considered, CRC patients with low *HSPD1* expression and advanced malignancy had a poorer prognosis than those with high *HSPD1* expression. Thus, *HSPD1*/HSP60 can be a useful biomarker for a sophisticated survival prediction in late- and advanced-stage CRC, allowing the design of individualized treatment strategies.

## 1. Introduction

Colorectal cancer (CRC) has the third-highest incidence rate and ranks second in causing mortality worldwide [1]. In countries with a high human development index, such as the United States, France, and Japan, its incidence is decreasing because of screening and prevention; however, its incidence is increasing in developing countries [2]. In the United States, where CRC incidence and mortality are declining, CRC incidence and mortality are increasing at a young age, unlike in old-age individuals [3,4]. With its increasing global burden, CRC remains an important health problem.

Targeted therapies have been developed owing to changes in the cancer treatment paradigm, thereby increasing cancer survival rates. Targeted therapeutics target the specific factors involved in cancer progression. Therefore, even in the same type of cancer, there is a limitation in that it may only be effective in some subgroups of patients with cancer in which a specific target factor is expressed. Ultimately, investigating novel biomarkers for tumor subgroups is the first step in targeted therapies. Biomarkers are specific targets not only for protein expression but also for targeting the stress-associated microenvironment.

Heat shock protein 60 (HSP60, also called Cpn60), a stress protein that acts as a defense mechanism in cells [5], is a promising candidate for the prevention and treatment of CRC. HSP60, a protein encoded by the heat shock protein family D member 1 (*HSPD1*) gene, protects cells by increasing their production when exposed to harmful external stimuli [6]. In other words, it prevents the abnormal misfolding or aggregation of proteins [7], aids in the process of protein migration to intracellular organelles, and breaks down denatured proteins [8]. HSP60 plays a role in tumor growth and progression in several cancers, including ovarian [9], gastric cancer [10], liver cancer [11], pancreas cancer [12], breast cancer [13], papillary thyroid carcinoma [14], cervical cancer [15], head and neck cancer [16], and CRC [17,18,19]. However, there are conflicting reports that increased HSP60 expression correlates with better oncological outcomes [20,21]. For example, in breast cancer, the higher the expression of HSP60, the worse the prognosis owing to increased metastasis [22], whereas in esophageal adenocarcinoma patients, the prognosis is good because of a positive response to platin/5-FU treatment [23]. Moreover, HSP60 overexpression in CRC tissues is associated with poor prognosis [24,25,26]. Therefore, the effect of the increased expression of HSP60 on cancer is conflicting, and further studies are needed to determine its clinical significance.

Although there have been some reports on the potential of HSP60 as a biomarker, very few studies have demonstrated its clinical efficacy, particularly in CRC. In the present study, we conducted a study to clarify the clinical significance of HSP60 in CRC, which is increasing worldwide. In addition, we performed an additional bioinformatics analysis using The Cancer Genome Atlas (TCGA) data on the survival of patients with CRC according to the expression level of *HSPD1*, which encodes the HSP60 protein. To provide more accurate prognostic prediction evidence, we analyzed TNM classification. Our results indicate that low HSP60 expression is associated with poor prognosis in advanced CRC, suggesting that HSP60 can enable more sophisticated prognosis prediction and personalized treatment strategies.

## 2. Materials and Methods

### 2.1. Definition of Clinicopathologic Factors

Family history of CRC was defined as the development of CRC in a first-degree relative. Patients who smoked in the past or were current smokers were defined as having a smoking factor. Anemia was defined as a hemoglobin level of <13 g/dL in men and <12 g/dL in women. Abnormal white blood cell (WBC) count was defined as follows: leukocytosis when the count is greater than 10,000 cells/μL or leukopenia when the count is lower than 4000 cells/μL. Abnormal carcinoembryonic antigen (CEA) level was defined as >5 ng/mL. CRC is characterized by the location of the primary tumor. Right-sided colorectal cancer (RCC) is defined as cancer of the cecum, ascending colon, or transverse colon up to the hepatic flexure. Left-sided colorectal cancer (LCC) is defined as cancer of the splenic flexure and regions distal to the splenic flexure, including the rectum.

Differentiated and undifferentiated cancers were classified according to the pathological findings that affected the differentiation and prognosis of CRC. Cancers with good differentiation (WD) and moderate differentiation (MD) were classified as differentiated cancers, and poorly differentiated (PD) cancers, signet ring cell (SRC) carcinoma, and mucinous carcinoma, which have a poor prognosis, were classified as undifferentiated cancers.

### 2.2. HSP60 by Tissue Samples

A tissue microarray (TMA) was performed on the tumor tissue of CRC patients to examine their survival rate according to the expression level of the HSP60 protein encoded by the *HSPD1* gene, and the HSP60 protein expression level was confirmed after immunohistochemistry (IHC) staining.

#### 2.2.1. Patient Population and Clinical Specimens

This study included patients who underwent CRC surgery at Gachon University Gil Medical Center between April 2010 and January 2013. A total of 456 patients with primary CRC who underwent surgery and had preserved paraffin blocks were analyzed. Patients with recurrent CRC, changes in the normal bowel structure of the colon as a result of previous surgery, a history of chemotherapy or abdominal radiation therapy prior to CRC surgery, and those treated for other cancers before CRC surgery were excluded. Study approval was obtained from the Institutional Review Board (GCIRB2023-099).

#### 2.2.2. Tissue Microarray (TMA) and IHC

After reviewing the H&E-stained CRC slides, two representative CRC cores were selected. Cylindrical formalin-fixed paraffin-embedded tissues (2 mm in diameter) were obtained using TMA. A total of 69 CRC cases were included in a paraffin-embedded TMA block. IHC for anti-HSP60 was performed to confirm HSP60 expression levels. TMA blocks were sectioned using microtome at about 4 µm thickness. Slides were baked at 60 °C for 10 min, deparaffinized with xylene, and rehydrated using an alcohol gradient (100% alcohol, 95% alcohol, 80% alcohol, and 70% alcohol). The tissue slides were then treated with 3% hydrogen peroxide in methanol for 10 min to quench endogenous peroxidase activity, and the antigens were retrieved in 0.01 M sodium citrate buffer (pH 6.0) using a microwave oven. After 30 min of preincubation in 10% normal goat serum to prevent nonspecific staining, the samples were incubated overnight using anti-HSP60 (1:200, #D6F1, monoclonal, Cell Signaling Technologies, Danvers, MA, USA) in a humidified container at 4 °C. The tissue slides were treated with a non-biotin horseradish peroxidase detection system according to the manufacturer’s instructions (Gene Tech, San Francisco, CA, USA). The degree of HSP60 expression was classified into four stages from 0 to 3. TMA. Two experienced pathologists (M.K. and J.A.) unbiasedly analyzed the TMA slides twice. The classification of the expression intensity of HSP60 using IHC was as follows: point 0 indicated no staining, point 1 indicated weak staining, point 2 indicated moderate staining, and point 3 indicated strong staining. For expression levels, 0–2 points were defined as low expression and 3 points as high expression (Figure 1).

#### 2.2.3. Statistical Analysis

Continuous variables are expressed as mean values with ranges, whereas categorical variables are shown as absolute numbers and percentages. The interdependence between HSP60 and clinical data was calculated using the chi-squared test. In this study, survival-related factors and survival rates focused on event-free survival (EFS) and disease-specific survival (DSS) at five years after CRC surgery. For EFS, events are defined as cancer progression, cancer recurrence, and death due to CRC. DSS was defined as the time until death due to CRC after surgery and was considered censored in cases of death from diseases other than CRC. The TNM classification and HSP60 protein levels were used to predict survival. Patients were divided into four groups (group 1: high HSP60 protein expression (hereafter referred to as HSP60_High_) and TNM classification stage I/II; group 2: low HSP60 protein expression (hereafter referred to as HSP60_Low_) and TNM classification stage I/II; group 3: HSP60_High_ and TNM classification stage III/IV; group 4: HSP60_Low_ and TNM classification stage III/IV). EFS and DSS were analyzed according to protein expression levels.

Survival curves were plotted using the Kaplan–Meier method with the log-rank test. Survival analysis was performed using the Cox proportional hazards regression. SPSS (version 22.0; SPSS Inc., Chicago, IL, USA) was used for the statistical analysis. Statistical significance was set at *p* < 0.05 (two-sided).

### 2.3. HSPD1 Statistical Analysis Utilizing the TCGA Data Set

Gene expression and clinical information datasets from the CRC datasets of TCGA project were downloaded from the UCSC Cancer Genomics Browser (last accessed on 3 December 2017, 19333237). Survival analysis was performed on 272 *HSPD1*-expression CRC samples and 272 samples with 5-year overall survival (OS) and relapse-free survival (RFS) data. The TCGA data had limitations in terms of confirmable information; therefore, relapse-free survival and OS were examined.

Survival rate analysis was performed using the log-rank test, a univariate analysis tool. MaxStat function2 in R version 4.0 was used to determine the optimal cut-off point for the expression levels of *HSPD1* [27]. For OS analysis, we set a higher expression group (*HSPD1*_High_) when the expression level of *HSPD1* was greater than 14.1791 and a lower expression group (*HSPD1*_Low_) when the expression level of *HSPD1* was less than or equal to 14.1791. For relapse-free survival analysis, we set a higher expressed group (*HSPD1*_High_) when the expression level of *HSPD1* was greater than 14.1211 and a lower expressed group (*HSPD1*_Low_) when the expression level of *HSPD1* was less than or equal to 14.1211.

### 2.4. Mutivariate Analysis

Multivariate analysis using the Cox proportional hazards (CPH) model was performed to account for age and sex. The survival package in R version 4.0 was used for the log-rank test and the CPH model for survival rate analysis. The forestmodel package was used with R version 4.0 to determine the HR for each variable acquired using the CPH model.

## 3. Results

### 3.1. The Association of HSP60 with Clinicopathological Variables

To elucidate the clinicopathological significance of HSP60 in CRC, we examined the IHC levels of HSP60 in CRC tissues using the TMA cohort (Figure 1). HSP60 was mainly located in the cytoplasm of the tumor cells (granular cytoplasmic expression).

The mean age of 456 patients who underwent the analysis was 65.0 ± 11.4 years (Table 1). The number of male patients was 272 (59.6%), and the mean tumor size was 51.8 ± 21.6 mm. Cancer of the rectum accounted for the majority of CRC cases, with 120 (26.3%) cases, and the degree of differentiation was the highest with 387 (84.9%) cases of MD cancer. The TNM classifications of CRC were as follows: 77 (16.9%) stage I, 163 (35.7%) stage II, 160 (35.1%) stage III, and 56 (12.3%) stage IV.

The clinical significance of the expression pattern of HSP60 is as follows (Table 2). In multivariate analysis using logistic regression in 456 patients with CRC, undifferentiated type (odds ratio [OR], 0.269; 95% CI: 0.127–0.571; *p* = 0.001) and p53 mutation type (OR, 1.662; 95% CI: 1.042–2.651; *p* = 0.033) were independent predictive risk factors for HSP60 high expression in CRC. However, the expression level of HSP60 was not significantly related to the following clinicopathological parameters: age, sex, diabetes mellitus, family history, smoking, anemia, WBCs count, serum CEA level, tumor location, TNM classification, or EGFR mutation.

### 3.2. CRC Prognostication Based on HSP60 Expression

#### 3.2.1. Analysis of Survival Rate according to HSP60 Expression Level

We performed a survival analysis to evaluate the association between HSP60 expression and 5-year EFS (Figure 2A). The HSP60_Low_ survival rate within the 5-year EFS was 67.4%, HSP60_High_ survival rate was 74.7% (*p* = 0.042), and median 5-year EFS for the HSP60_Low_ and HSP60_High_ groups was 1406.6 days and 1551.1 days, respectively. In addition, the HR of HSP60_Low_ group vs. and HSP60_High_ groups was 1.42 in the multivariate model (95% CI: 1.01–2.00, *p* = 0.04).

Furthermore, HSP60_Low_ and HSP60_High_ survival rates within the 5-year DSS were 68.8% and 79.3% (*p* = 0.013), and median 5-year DSS for HSP60_Low_ and HSP60_High_ groups was 1451.7 days and 1620.8 days, respectively. Notably, the HR of 5-year DSS in the HSP60_Low_ was 1.69 (95% CI: 1.17–2.44, *p* = 0.005) (Figure 2B), suggesting that high HSP60 expression is associated with better survival than low HSP60 expression.

#### 3.2.2. Survival Prediction of CRC Patients Combining TNM Classification and HSP60 Expression Level

When 5-year EFS was assessed by combining HSP60 and TNM classifications, the HRs of EFS in groups 3 and 4 compared with group 1 were 4.18 (95% CI: 2.33–7.49, *p* < 0.001) and 7.21 (95% CI: 4.05–12.83, *p* < 0.001), respectively (Figure 3A). In particular, the 5-year EFS HR for the late/advanced stage (TNM classification stage III/IV) in the HSP60_Low_ group was 1.70 (95% CI: 1.14–2.54, *p* = 0.009), indicating a significantly worse prognosis than the HSP60_High_ group (Figure 3C). These findings imply that combining HSP60 and TNM classification can improve survival prediction compared to TNM classification alone.

When HSP60 and TNM classifications were integrated and examined in the DSS analysis, compared with group 1, the HR of DSS in group 3 was 4.52 (95% CI: 2.31–8.83, *p* < 0.001), and group 1 versus group 4 was 9.40 (95% CI: 4.89–18.07, *p* < 0.001) (Figure 3B). In patients with the same stage of disease (TNM classification stage III/IV), the HR of DSS in the HSP60_Low_ group was 2.02 (95% CI: 1.33–3.08, *p* = 0.001), indicating a significantly worse prognosis than the HSP60_High_ group (Figure 3D).

### 3.3. HSPD1 and Colorectal Cancer Patients’ Survival

#### 3.3.1. Correlation between HSPD1 Expression Level and Survival Rate

Similar to HSP60, the 5-year RFS and 5-year OS were analyzed to determine whether there was a difference in survival rate depending on the expression of *HSPD1*, which encodes HSP60. Patients with high *HSPD1* expression had an advantage in terms of survival compared with those with low expression levels (Figure 4A,B).

#### 3.3.2. Survival Prediction of CRC Patients Combining TNM Classification and HSPD1 Expression Level

Compared with group 1, the HR of 5-year RFS was 1.10 (95% CI: 0.40–3.04, *p* = 0.854) in group 2, 1.45 (95% CI: 0.44–4.75 *p* = 0.541) in group 3, and 3.76 (95% CI: 1.49–9.53, *p* = 0.005) in group 4 (Figure 5A). Furthermore, when the HR of 5-year OS was evaluated by combining *HSPD1* and TNM classifications, compared with group 1, the HR in group 2 was 1.04 (95% CI: 0.35–3.10, *p* = 0.940), for group 3, it was 2.02 (95% CI: 0.61–6.69, *p* = 0.248), and for group 4, it was 5.00 (95% CI: 1.85–13.52, *p* = 0.002) (Figure 5B). Interestingly, the HR of 5-year RFS and 5-year OS risk for group 4, HSPD1_Low_ in patients with late/advanced stage (TNM classification stage Ⅲ/Ⅳ), was 2.48 (95% CI: 0.89–6.89, *p* = 0.08) and, 2.21 (95% CI: 0.89–5.48, *p* = 0.09), respectively, indicating a tendency toward poorer prognosis than the *HSPD1*_High_ group (group 3) (Figure 5C,D).

## 4. Discussion

This study demonstrated the important role of *HSPD1*/HSP60 in predicting the prognosis of advanced CRC. The higher the *HSPD1*/HSP60 expression, the better the prognosis, which varied considerably depending on the degree of *HSPD1*/HSP60 expression, especially in TNM stages III/IV.

One of the major features of this study was that TNM classification was also considered in predicting CRC prognosis. TNM classification has long been the most common method for providing clinical practice diagnosis/treatment/prognosis assessment information, and the stages are classified into stages I–IV. Because the TNM classification is simplistic and does not contain all significant characteristics that influence prognosis, prognosis prediction using the TNM classification alone is incomplete. Furthermore, there are no previously known markers that can be used in combination with the TNM classification of data. According to the results of this study, HSP60 alone has relevance as a predictor of prognosis at the late/advanced stages, and the higher the HSP60, the better the prognosis, indicating it acts as an ‘anti-tumor marker’. Previous studies have indicated that the higher the T and N stage, the poorer the prognosis of CRC [28,29]. However, combining HSP60 expression with individualized treatment may be more beneficial, as our analysis of clinicopathologic significance in HSP60 expression indicates that it is independent of the TNM classification.

HSPs are important proteins involved in protein folding and have numerous subtypes [30]. Although there are several studies on the importance of HSP60, only a few of them are on HSP60 expression in cancer. Furthermore, previous research has elucidated the role of HSP60 in in vitro or in vivo models of almost all cancers [24,31,32,33,34]. Additionally, the clinical significance has been evaluated through differences in HSP60 expression in serum and tissue between normal individuals and patients with cancer using enzyme-linked immunosorbent assays, IHC, Western blotting, and quantitative reverse transcription-polymerase chain reactions [13,35,36,37,38]. However, the relationship between HSP60 expression and survival has not yet been clarified. HSP60 expression in serum and tissue serves as indirect evidence of CRC prognosis. In this study, we determined the level of HSP60 expression in tissues of patients with CRC, performed patient survival analysis to determine its clinical significance, and directly assessed CRC prognosis through HSP60 expression. Furthermore, a quick and low-cost diagnostic method is preferred over a method that is expensive and takes a long time to diagnose in real-world settings. IHC has long been used as a low-cost method for cancer diagnosis, with abundant relevant information. Therefore, survival analysis performed after categorizing HSP60 expression intensity using IHC can provide important information to medical staff and patients for treating patients in real-world settings.

HSP60 can play a critical role in inhibiting tumor progression and promoting apoptosis by interacting with various proteins, particularly p53 [39]. Upon receiving stress signals from either external factors or within the cells, HSP60 undergoes acetylation. This modification leads to an increase in the free p53 levels, which in turn activates the p53-dependent pathway, ultimately facilitating tumor cell death via proteins like Bax-Cytochrome c [6,39]. In this study, the HSP60_High_ group showed a statistically significant increase in p53 expression (*p* = 0.033 in multivariate analysis). Reduced p53 expression in the HSP60_Low_ group possibly inhibits the interaction between HSP60 and p53 and subsequently promotes tumor formation, resulting in a lower survival rate than that in the HSP60_High_ group.

Unlike the findings of our study that directly verify the clinical significance of HSP60 in patient tissues, previous studies performed with body fluids including blood, have shown increased HSP60 levels in the serum of patients with advanced-stage CRC. Furthermore, serum carcinoembryonic antigen (CEA), commonly used for CRC diagnosis or recurrence [40], has been reported to have a positive correlation with serum HSP60 levels [41]. Although serum HSP60 levels were not assessed in this study, the comparison of serum CEA levels with the HSP60 expression in tissues was not statistically significant. We confirmed the potential use of an independent marker to predict CRC prognosis in the absence of other factors.

Based on the findings of previous in vivo and in vitro studies, HSP60 plays a dual role in tumor suppression and tumor generation; however, we found that low HSP60 expression of advanced-stage CRC is associated with poor prognosis, and the substantial association between HSP60 and p53 expression provides additional evidence to support this finding. Despite the fact that this was a single-institution study, we were able to obtain many cases. The importance of this study is that it offers more precise prognostic predictors in addition to TNM classification using IHC, which may be economically applicable in the clinical field.

## 5. Conclusions

This study highlighted *HSPD1*/HSP60 as biomarkers for predicting CRC prognosis. *HSPD1*/HSP60 expression levels can more precisely predict the prognosis of CRC when combined with TNM classification. This enables more sophisticated forecasting and the advanced development of personalized strategies (Figure 6).

## Figures and Tables

**Figure 1 cancers-15-04052-f001:**
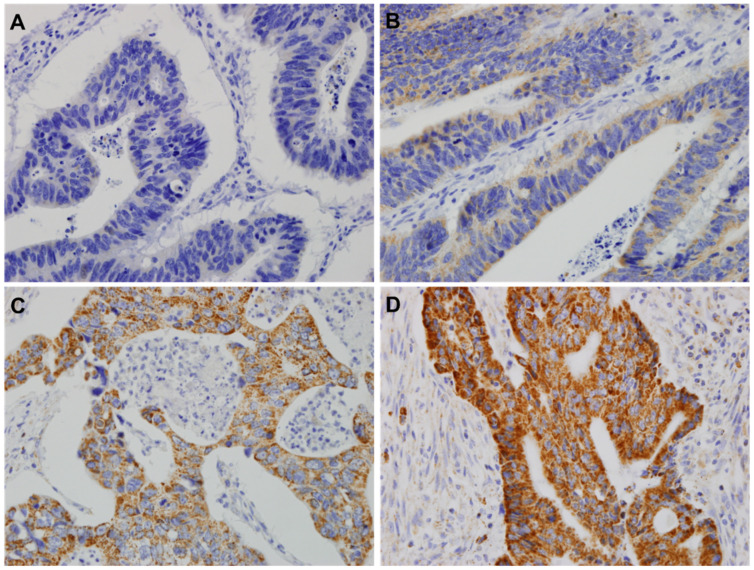
Protein expression in colorectal cancer (CRC) tissue, as analyzed via immunohistochemistry. (**A**) Score of 0 showing no staining. (**B**) Score of 1+ showing faint and weak cytoplasmic staining. (**C**) Score of 2+ showing moderate smooth granular cytoplasmic staining. (**D**) Score of 3+ showing strong granular cytoplasmic staining. Original magnification, 400×. HSP60, heat shock protein 60.

**Figure 2 cancers-15-04052-f002:**
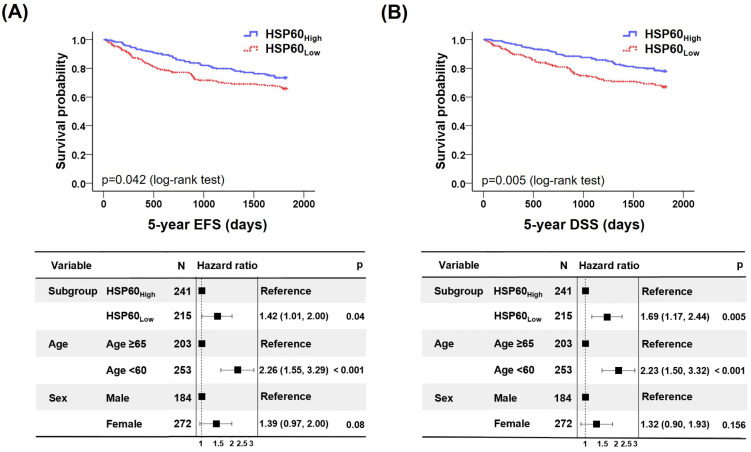
The lower the expression of HSP60 in CRC patients, the worse the prognosis. (**A**,**B**) Event-free survival (EFS) (**A**) and disease-specific survival (DSS) (**B**) of patients with low (red line) and high (blue line) expression of HSP60.

**Figure 3 cancers-15-04052-f003:**
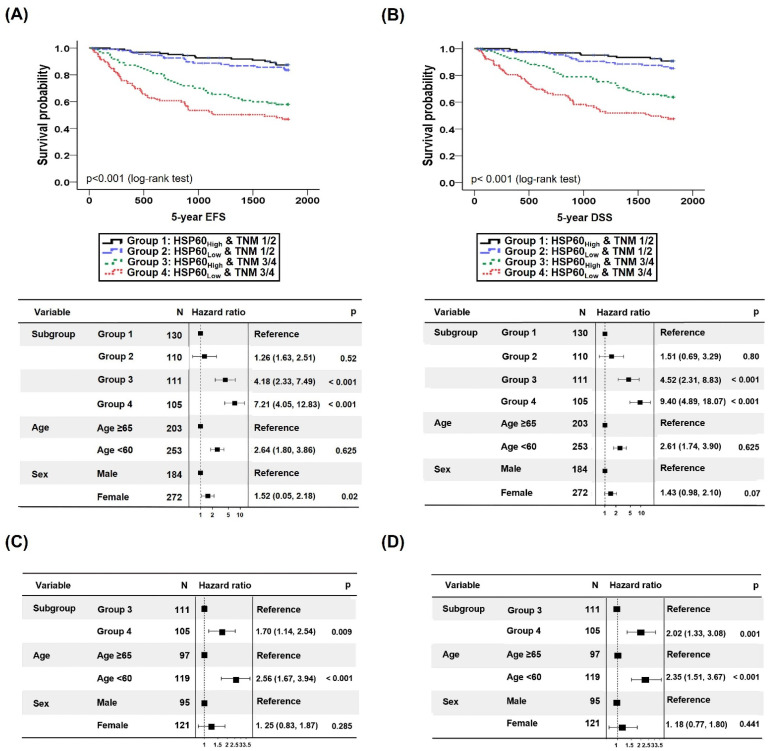
The higher the HSP60 expression in patients with advanced CRC, the lower the survival hazard ratio (HR). (**A**,**B**) EFS (**A**) and DSS (**B**) curves of CRC patients according to TNM classification and HSP60 expression. (**C**,**D**) HRs of EFS (**C**) and DSS (**D**) in CRC patients based on HSP60 expression at late stage.

**Figure 4 cancers-15-04052-f004:**
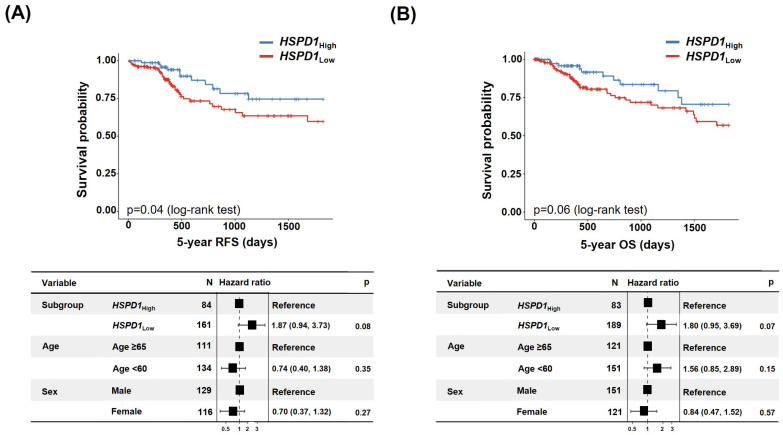
Reduced *HSPD1* expression is associated with a poor prognosis. (**A**,**B**) relapse-free survival (RFS) (**A**) and Overall survival (OS) (**B**) of CRC patients based on *HSPD1* levels.

**Figure 5 cancers-15-04052-f005:**
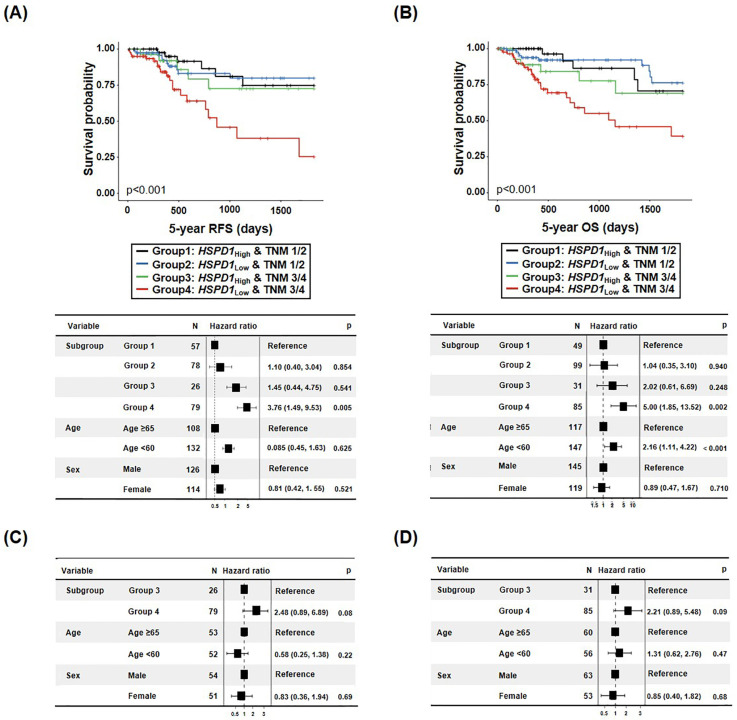
The lower the *HSPD1* expression in patients with late CRC, the worse the survival rate. (**A**,**B**) CRC patients’ RFS (**A**) and OS (**B**) curves based on TNM classification and *HSPD1* expression. (**C**,**D**) CRC patients’ risk rate according to advanced stage *HSPD1* levels.

**Figure 6 cancers-15-04052-f006:**
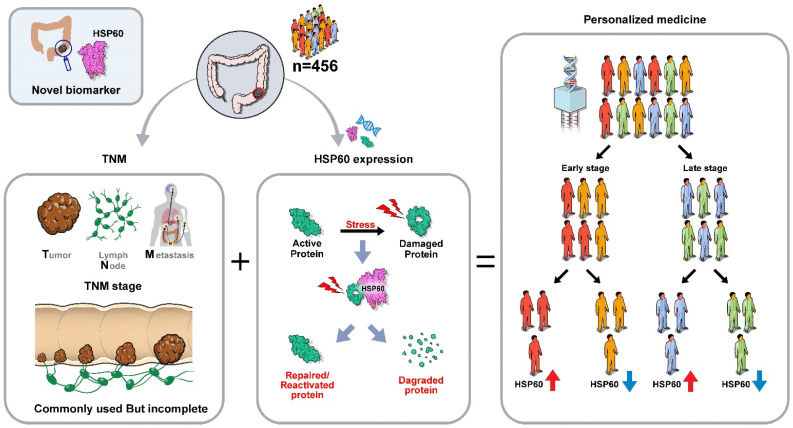
Schematic representation of developing personalized medicine based on HSP60 and TNM stage.

**Table 1 cancers-15-04052-t001:** Baseline characteristics.

	Total (*n* = 456)
Age	65.0 ± 11.4
Sex	
Male	272 (59.6%)
Female	184 (40.4%)
Laboratory findings	
Hemoglobin (g/dL)	12.3 ± 2.4
WBC (10^3^/μL)	11,747.9 ± 4031.8
CEA (ng/mL)	80.7 ± 405.2
Histology	
Size (mm)	51.8 ± 21.6
Location	
Cecum	9 (2.0%)
Ascending colon	79 (17.3%)
Hepatic flexure	8 (1.8%)
Transverse colon	28 (6.1%)
Splenic flexure	4 (0.9%)
Descending colon	12 (2.6%)
Sigmoid-descending	5 (1.1%)
Sigmoid colon	110 (24.1%)
Rectosigmoid colon	81 (17.8%)
Rectum	120 (26.3%)
Pathology	
WD	29 (6.4%)
MD	387 (84.9%)
PD	20 (4.4%)
Mucinous	18 (3.9%)
SRC	2 (0.4%)
TNM stage	
Ⅰ	77 (16.9%)
Ⅱ	163 (35.7%)
Ⅲ	160 (35.1%)
Ⅳ	56 (12.3%)

WBC, white blood cell; CEA, carcinoembryonic antigen; WD, well differentiated, MD, moderately differentiated; PD, poorly differentiated; TNM, tumor node metastasis; SRC, signet ring cell.

**Table 2 cancers-15-04052-t002:** Clinicopathologic significance of HSP60 expression.

Variables		Univariate			Multivariate	
	Total (*n* = 456)	HSP60 (*n* = 215), Low	HSP60 (*n* = 241), High	*p*-Value	OR (CI)	*p*-Value
Age (yrs)				0.609		
<65	203 (44.5%)	93 (43.3%)	110 (45.6%)			
≥65	253 (55.5%)	122 (56.7%)	131 (54.4%)			
Sex				0.473		
Male	272 (59.6%)	132 (61.4%)	140 (58.1%)			
Female	184 (40.4%)	83 (38.6%)	101 (41.9%)			
Diabetes mellitus				0.735		
No	366 (80.3%)	174 (80.9%)	192 (79.7%)			
Yes	90 (19.7%)	41 (19.1%)	49 (20.3%)			
Smoking				0.983		
No	354 (77.6%)	167 (77.7%)	187 (77.6%)			
Yes	102 (22.4%)	48 (22.3%)	54 (22.4%)			
Family history				0.625		
No	437 (95.8%)	205 (95.3%)	231 (96.3%)			
Yes	19 (4.2%)	10 (4.7%)	9 (3.7%)			
Anemia (Hemoglobin, g/dL)				0.249		
No	233 (51.1%)	116 (54.0%)	117 (48.5%)			
Yes	223 (48.9%)	99 (46.0%)	124 (51.5%)			
WBC counts (10^3^/μL)				0.693		
Normal	385 (84.4%)	180 (83.7%)	205 (85.1%)			
Abnormal	71 (15.6%)	35 (16.3%)	36 (14.9%)			
Serum CEA (ng/mL)				0.653		
Normal	337 (73.9%)	161 (74.9%)	176 (73.0%)			
Abnormal	119 (26.1%)	54 (25.1%)	65 (27.0%)			
Tumor size (mm)				0.539		
<50	220 (48.2%)	107 (49.8%)	113 (46.9%)			
≥50	236 (51.8%)	108 (50.2%)	128 (53.1%)			
Tumor location				0.044		0.164
LCC	332 (72.8%)	147 (68.4%)	185 (76.8%)		1	
RCC	124 (23.2%)	68 (31.6%)	56 (23.2%)		0.732 (0.477–1.124)	
Tumor differentiation				<0.001		0.001
Differentiation	416 (91.2%)	185 (86.0%)	231 (95.9%)		1	
Undifferentiation	40 (8.8%)	30 (14.0%)	10 (4.1%)		0.269 (0.127–0.571)	
p53 expression ^a^				0.045		0.033
Negative	93 (20.7%)	52 (24.8%)	41 (17.1%)		1	
Positive	357 (79.3%)	158 (75.2%)	199 (82.9%)		1.662 (1.042–2.651)	
EGFR mutation ^b^				0.074		
Negative	306 (64.2%)	152 (72.0%)	154 (64.2%)			
Positive	145 (32.2%)	59 (28.0%)	86 (35.8%)			
TNM stage				0.553		
Ⅰ/Ⅱ	240 (52.6%)	110 (51.2%)	130 (53.9%)			
Ⅲ/Ⅳ	216 (47.4%)	105 (48.8%)	111 (46.1%)			

OR, odds ratio; CI, confidence interval; ^a^ P53 mutation could not be evaluated in 6 cases. ^b^ EGFR mutation could not be evaluated in 5 cases.

## Data Availability

The data presented in this study are available in this article.

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
