# Peer review of "Clinicopathologic Significance of Heat Shock Protein 60 as a Survival Predictor in Colorectal Cancer"

_cancers, 2023, doi:10.3390/cancers15164052_

Round 1

Reviewer 1 Report

THE PRESENT MANUSCRIPT EXPLAIN THE CLINICOPATHOLOGIC SIGNIFICANCE OF HEAT SHOCK PROTEIN AS A SURVIVAL PREDICTOR IN COLORECTAL CANCER. NEED OF THE PRESENT INVESTIGATION IS EXPLAINED WELL AND THE OBJECTIVE IS CLEARLY MENTIONED. THE METHODS ADOPTED WAS FOUND TO BE APPROPRIATE AND REPRODUCIBLE. IT IS HIGHLY APPRECIABLE THAT THE AUTHORS MENTION THE ETHICAL CLEARANCE NUMBER IN THE MANUSCRIPT. THE RESULTS ARE PRESENTED WELL AND EXPLAINED WELL. IF THE CLARITY OF THE FIGURES ARE IMPROVED, IT WILL BE USEFUL FOR THE READERS. THE RESULTS ARE DISCUSSED WELL AND IT CLEARLY MENTION THE STRENGTH AND WEAKNESS OF THE PRESENT STUDY. THE FLOW OF THE MANUSCRIPT WAS ALSO FOUND TO BE GOOD.

Author Response

We thank the reviewer for the positive feedback and valuable comment. Following the reviewer’s comment, we have explained the clinicopathological importance of HSP60 as a survival predictor in patients with colorectal cancer by confirming the survival rate of patients with colorectal cancer by considering both HSP60 expression and TNM stage. The results of this study will assist in the personalized treatment of patients with colorectal cancer. Furthermore, to help readers better understand our results, we have revised the manuscript (as marked in red font in the revised version) and replaced the figures with higher-quality ones.

Reviewer 2 Report

There have been many clinical studies on HSP60, and there have been many discussions as a predictor of prognosis. It's hard to highlight the value in this article. Although there is a more comprehensive bioinformatics analysis, the clinical application value is low.

Author Response

We thank the reviewer for the valuable comment, which contributed to improving the quality of this research. As you mentioned, there are reports on the role of HSP60 as a predictor of cancer prognosis. Several studies on HSP60 are available on Pubmed, but most of them are related to diseases other than cancer than cancer (such as cardiovascular diseasea, b, rheumatoid arthritisc, d) and its role as a chaperone, f in cells. There are only a few studies on the role of HSP60 in cancer. In particular, only a few studies have determined the role of HSP60 in the clinical aspects of colorectal cancer. In addition, most of these studies have been conducted in vitrog, h, I, j and in vivoj, k, and have confirmed only the difference in HSP60 expression in serum or tissue between normal individuals and patients with colorectal cancer using enzyme-linked immunosorbent assayl, m, immunohistochemistrym, n, o, p, q, and quantitative reverse transcription-polymerase chain reactiono, r. The results of these studies can only indirectly explain the clinical significance of HSP60. Moreover, only 2–3 studies have evaluated HSP60 considering clinical variables. In this study, we investigated HSP60 expression in tissues from patients with colorectal cancer and validated its association with overall survival and recurrence survival. Furthermore, the possibility of HSP60 as a predictor of colorectal cancer prognosis was directly confirmed, as the patient survival rate was verified by considering the expression of HSP60 and TNM stage, the most commonly used variables in cancer diagnosis.

To further clarify that our findings are important and valuable clinically, the clinical significance of HSP60 was strengthened by adding the above contents in the revised manuscript (as marked in red font in the revised version).

Ref:

a) Role of Heat Shock Proteins in Atrial Fibrillation: From Molecular Mechanisms to Diagnostic and Therapeutic Opportunities. Cells. 2020 Dec 30;12(1):151.

b) Emerging role of heat shock proteins in cardiovascular diseases. Adv Protein Chem Struct Biol. 2023 Feb;134:271-306.

c) A peptide derived from HSP60 reduces proinflammatory cytokines and soluble mediators: a therapeutic approach to inflammation. Front Immunol. 2023 Apr 28;14:1162739.

d) Targeting of tolerogenic dendritic cells to heat-shock proteins in inflammatory arthritis. J Transl Med. 2019 Nov 14;17(1):375

e) Crystal structures of dimeric and heptameric mtHSP60 reveal the mechanism of chaperonin inactivation. Life Sci Alliance. 2034 Mar 27;6(6):e202201753

f) Heat-Shock Proteins. Curr Protoc. 2022 Nov;2(11):e592

g) HSP60-knockdown suppresses proliferation in colorectal cancer cells via activating the adenine/AMPK/mTOR signaling pathway. Oncol Lett. 2021 Aug;22(2):630.

h) Exploiting the HSP60/10 chaperonin system as a chemotherapeutic target for colorectal cancer. Bioorg Med Chem. 2021 Jun 15;40:116129.

i) Fhit-Fdxr interaction in the mitochondria: modulation of reactive oxygen species generation and apoptosis in cancer cells. Cell Death Dis. 2019 Feb 15;10(3):147.

j) UBXN2A enhances CHIP-mediated proteasomal degradation of oncoprotein mortalin-2 in cancer cells. Mol Oncol. 2018 Oct;12(10):1753-1777.

k) Effective photoimmunotherapy of murine colon carcinoma induced by the combination of photodynamic therapy and dendritic cells. Clin Cancer Res.2004 Jul 1;10(13):4498-508.

l) Novel serum markers HSP60, CHI3L1, and IGFBP-2 in metastatic colorectal cancer. Oncol Lett. 2019 Dec;18(6):6284-6292. 

m) Identification and verification of heat shock protein 60 as a potential serum marker for colorectal cancer. FEBS J. 2011 Dec;278(24):4845-59.

n) Heat shock protein 60 levels in tissue and circulating exosomes in human large bowel cancer before and after ablative surgery. Cancer. 2015 Sep 15;121(18):3230-9.

o) Differential expression of Janus kinase 3 (JAK3), matrix metalloproteinase 13 (MMP13), heat shock protein 60 (HSP60), and mouse double minute 2 (MDM2) in human colorectal cancer progression using human cancer cDNA microarrays. Pathol Res Pract. 2005 Dec;201(12):777-89.

p) The expression of HSP60 and HSP10 in large bowel carcinomas with lymph node metastase. BMC Cancer. 2005 Oct 28;5:139.

q) 60KDa chaperonin (HSP60) is over-expressed during colorectal carcinogenesis. Eur J Histochem. 2003;47(2):105-10.

r) [Expressions of heat shock protein (HSP) family HSP 60, 70 and 90alpha in colorectal cancer tissues and their correlations to pathohistological characteristics]. Ai Zheng. 2009 Jun;28(6):612-8.

Reviewer 3 Report

The authors have prepared a well-described MS that offers a new observation to the field of CRC biomarkers. 

The only suggestions I have is for them to describe the recommendation(s) for HSP60 antibodies to use for biomarker analysis(for the general use by others)

AND do HSP60 levels in blood become elevated in CRC patients with high or low prognosis?  Address or at least use the literature to speculate. 

Author Response

We thank the reviewer for appreciating our study.

When diagnosing patients in a real-world setting, a low-cost and quick process is preferred over an expensive and time-consuming method. IHC, which we used to predict prognosis, in patients, is a commonly used method in clinical settings. It can quickly and inexpensively evaluate the intensity of HSP60 expression in patients and classify them by TNM stages. This assay can provide important information for patient treatment, and its potential clinical application has been determined. The Discussion has been revised to highlight these clinical applications (page 12, line 16).

A previous study has reported an increase in the serum level of HSP60, but its correlation with disease prognosis has not been confirmed. We directly confirmed the expression of HSP60 in cancer tissue rather than using an indirect method such as determining blood HSP60 levels. Although more studies are needed, our findings can help predict disease prognosis according to the level of HSP60 expression in blood.